# Early Detection of Late Onset Sepsis in Extremely Preterm Infants Using Machine Learning: Towards an Early Warning System

Arno G. Garstman [1,*], Cristian Rodriguez Rivero [2,*] and Wes Onland [3,4,*]

1 Institute of Informatics, University of Amsterdam, 1098 XH Amsterdam, The Netherlands
2 Centre for Research in Intelligent Sensors and Systems, Cardiff Metropolitan University, Cardiff CF52YB, UK
3 Department of Neonatology, Emma Children's Hospital, Amsterdam UMC, University of Amsterdam, 1105 AZ Amsterdam, The Netherlands
4 Amsterdam Reproduction & Development (AR&D) Research Institute, Amsterdam UMC, 1105 AZ Amsterdam, The Netherlands
* Correspondence: agarstman@icloud.com (A.G.G.); crodriguezrivero@cardiffmet.ac.uk (C.R.R.); w.onland@amsterdamumc.nl (W.O.)

**Abstract:** A significant proportion of babies that are admitted to the neonatal intensive care unit (NICU) suffer from late onset sepsis (LOS). In order to prevent mortality and morbidity, the early detection of LOS is of the utmost importance. Recent works have found that the use of machine learning techniques might help detect LOS at an early stage. Some works have shown that linear methods (i.e., logistic regression) display a superior performance when predicting LOS. Nevertheless, as research on this topic is still in an early phase, it has not been ruled out that non-linear machine learning (ML) techniques can improve the predictive performance. Moreover, few studies have assessed the effect of parameters other than heart rate variability (HRV). Therefore, the current study investigates the effect of non-linear methods and assesses whether other vital parameters such as respiratory rate, perfusion index, and oxygen saturation could be of added value when predicting LOS. In contrast with the findings in the literature, it was found that non-linear methods showed a superior performance compared with linear models. In particular, it was found that random forest performed best (AUROC: 0.973), 24% better than logistic regression (AUROC: 0.782). Nevertheless, logistic regression was found to perform similarly to some non-linear models when trained with a short training window. Furthermore, when also taking training time into account, K-Nearest Neighbors was found to be the most beneficial (AUROC: 0.950). In line with the literature, we found that training the models on HRV features yielded the best results. Lastly, the results revealed that non-linear methods demonstrated a superior performance compared with linear methods when adding respiratory features to the HRV feature set, which ensured the greatest improvement in terms of AUROC score.

**Keywords:** heart rate variability; respiratory frequency; perfusion index; late onset sepsis; premature infants; neonates; predictive monitoring; machine learning

## 1. Introduction

Approximately 11% of all babies are born preterm, of which 15.6% are born before 32 weeks of gestation and are admitted to the neonatal intensive care unit (NICU). These preterm infants are extremely vulnerable, with a high incidence of mortality and morbidity, especially due to late onset sepsis (LOS), which mostly occurs beyond the third day of life [1]. Prevention and early detection of this life-threatening bloodstream infection is of the utmost importance, particularly because this condition is not only associated with long-term morbidity and mortality, but it also results in increased costs resoling from prolonged hospitalization [2]. Directly after birth, preterm infants are continuously monitored and

often need intensive and prolonged treatment to support their vital functions. However, early signs of infectious episodes are non-specific and lead to both over- and undertreatment with antibiotics, which may disrupt normal microbiome development and can lead to antibiotic resistance [2]. Moreover, even when there is a clear indication of LOS and blood cultures are consequently taken, it can take up to 48 h before a result is available, which means that valuable time is lost when treating sepsis. Lastly, the results of blood cultures are prone to false positives due to contamination [3,4]. Therefore, a real-time non-invasive continuous monitoring system using an algorithm detecting a decrease in heart rate variability might be a useful early warning system for LOS in preterm infants. Currently, some hospitals are using such systems as a non-invasive tool to identify several threatening conditions at an early stage. One example of such a system is the heart rate observation system (HeRO). Among other conditions (e.g., necrotizing enterocolitis, meningitis, respiratory decomposition, brain pathology, and death), this system is capable of identifying several LOS prior to patient deterioration [2]. According to studies on the clinical effectiveness of this system, it has been shown that the HeRO monitor effectuated a significant decrease in the amount of mortality (2%) [5] and length of stay (3.2 days) [6]. Although these monitors have been shown to be quite effective, the algorithm is based on a linear model (i.e., logistic regression) [2], which might ignore possible non-linearities that are present in the data. Because of the complexity of LOS, it has been hypothesized that capturing these non-linearities with more advanced models might improve the LOS detection performance of the algorithm. Moreover, these monitors only use features derived from heart rate, and do not use other vital parameters such as perfusion index and oxygen saturation. Interestingly, a recent study using various ML methods found that logistic regression is quite robust and, in some cases, even superior with respect to some non-linear methods when predicting LOS [7,8]. However, the current literature is very scarce with respect to the assessment of non-linear methods when predicting LOS. For instance, adaptive boosting has not been assessed yet in the context of early LOS prediction, only in EOS (i.e., early onset sepsis) prediction [9]. Moreover, most studies have focused on ML-based methods to predict sepsis in adults and not in neonates [10]. Thus, although previous research has pointed out that mainly linear machine learning models are superior when predicting LOS, we hypothesized that some non-linear models could be potentially valuable when predicting non-linear dependencies in the data. Lastly, previous studies have not used all of the parameters that are available in NICU monitoring, as features derived from heart rate and respiratory variability are considered to be most valuable when predicting LOS [7,11–14]. To the best of our knowledge, perfusion index (PFI) and oxygen saturation (SpO2) have not yet been assessed in the context of LOS prediction. Thus, this study aims to assess the effect of using several non-linear ML methods, using features derived from vital parameters on the performance of LOS prediction in preterm infants. To do so, the following research questions are assessed:

- To what extent is the predictive performance of non-linear supervised ML models superior compared with linear supervised ML models when predicting LOS in preterm infants?
- What vital parameters, other than heart rate, are of added value when predicting LOS?

In the following sections, we first discuss the related works, where we describe the models and features that are considered valuable in the literature. Second, we describe the pre-processing of the data and the different variants of features and models that are used. Finally, we present an evaluation of the models used on different variants of the feature set and compare them with other results in the literature.

## 2. Literature Review

### 2.1. Models

The use of non-linear ML methods when predicting LOS in neonates is still a relatively novel and undiscovered subject within the literature. One of the reasons for this is that studies have focused on sepsis prediction in adults more than in neonates [10,15,16]. Inter-

estingly, when both linear (e.g., logistic regression) and non-linear ML methods (e.g., random forest and support vector machines) were assessed for a cohort of LOS and non-LOS patients, it was found that logistic regression displayed a superior performance when predicting LOS [8]. Furthermore, Cabrera et al. (2021) [7] also showed that logistic regression performed best compared with several other ML models (Naive Bayes and k-nearest neighbors). This finding explains the success of the commercial HeRO tool, which also uses the logistic regression model [2]. As mentioned, this system has been shown to reduce the amount of neonatal mortality and has successfully been deployed in some hospitals as a noninvasive tool monitoring tool. Nevertheless, the results of the above-mentioned studies [7,8] should be interpreted with caution due to the small sample sizes used, which negatively affect the overall generalizability. Moreover, a study that investigated the effect of non-linear methods when predicting EOS (i.e., early onset sepsis) found evidence that ensemble methods (i.e., adaptive boosting) were the mist successful predictors of EOS [9]. Thus, given the fact that EOS and LOS are similar phenomena, one could argue that the use of ensemble methods (i.e., which are inherently non-linear) might also be promising when predicting LOS.

Moreover, as LOS is related to many features derived from various parameters (e.g., HRV and respiratory rate), it is highly likely that non-linearities in the data exist. Within a binary classification problem, this means that the data points are not linearly separable in the two categories (i.e., LOS and non-LOS). As a result, models that aim to classify these data points in two categories require non-linear properties to sufficiently discriminate between classes. For instance, one model that can capture non-linearities within data is the random forest model, as it uses various trees to categorize the data, instead of using a linear decision boundary such as logistic regression. In total, eight machine learning methods are assessed: logistic regression (LogR), naive Bayes (NB), decision trees (DT), random forest (RandF), K-nearest neighbors (KNN), support vector machine (SVM), adaptive boosting (ADA), and gradient boosting (GB). As mentioned, LogR is considered as the baseline model as it is generally considered to be the best performing model in the literature [7,8,17]. The models can be roughly categorized into three groups: linear ML methods (LogR and NB), conventional ML methods (NB, DT, KNN, and SVM), and ensemble ML methods (RandF, ADA, and GB). The last two groups, conventional and ensemble ML methods, are both considered non-linear models.

Table 1 presents an overview of all of the ML methods that have been assessed in the literature. KNN is the simplest model [18]. This model assumes that similar data points are near to each other. Based on this assumption, the KNN model classifies the data based on its k-nearest data points (i.e., neighbors). Furthermore, the SVM model classifies the data by fitting a hyperplane to the data [19]. If the data are not separable in the given feature space by the hyperplane, SVM performs a so-called "kernel trick", where the data are transformed to a feature space in which they are separable again. Both KNN and SVM have been shown to perform relatively robustly in medical settings [20,21]. Lastly, the DT model predicts the value of a target variable by learning simple decision rules inferred from the data features. These rules are mostly decided on by the Gini Impurity metric. This metric indicates the probability of misclassifying an observation given a certain split within the tree. Subsequently, the DT model decides its splits (i.e., nodes) upon the split with the lowest Gini value. The greatest advantage of the DT model is that it uses a "white box" model: this allows the user to fully interpret the decisions of the model. As a result, the DT model is considered to be very useful within a medical context [22]. However, a great disadvantage of DT is that it typically strongly overfits the training data, which makes the model less robust to unseen observations (i.e., the test data).

**Table 1.** Overview of all ML models that have been assessed by recent literature.

|  | LogR | NB | DT | RandF | KNN | SVM | ADA | GB |
|---|---|---|---|---|---|---|---|---|
| Leon et al. (2020) [8] | X |  | X | X | X | X |  |  |
| Joshi et al. (2020) [17] | X |  |  |  |  |  |  |  |
| Cabrera et al. (2021) [7] | X | X |  |  |  | X |  |  |
| Gomez et al. * (2019) [9] | X | X | X | X | X | X | X |  |

* this work assessed EOS instead of LOS.

The overview in Table 1 indicates that it is mainly the ensemble methods of ADA and GB that have not yet been assessed. Ensemble methods use multiple models (i.e., "learners"), typically decision trees. These models base their predictions on the majority predictions of each individual model within the ensemble, so-called "voting". For instance, RandF is an ensemble method that is constructed by a multitude of decision trees at the training time [23]. The predictive performance of the ensemble models is often better than what would be obtained from a single model alone. However, the interpretability of ensemble methods is often poor when compared with conventional methods, resulting in a "black box" model. Furthermore, ensemble models are often less prone to overfit towards the training data, as the multiple models within the ensemble (i.e., "learners") make them more robust towards unseen data.

Although RandF, ADA, and GB all rely on the principle of multiple learners, there are some key differences. For instance, while RandF trains its learners on a randomly selected subset of the data (i.e., "bagging"), ADA includes all features in each decision tree. As a result, ADA is more affected by noise (e.g., unimportant features) [24]. Moreover, ADA includes the principle of "boosting". Boosting relies on the principle that during each learning iteration, the learners (i.e., decision trees) that perform bad incrementally receive less votes. This way, the learners that are performing well become more important with respect to the predictions, which typically results in a higher accuracy. Another important difference between the two models is that ADA only uses a depth of one (i.e., two leaves) in each of its decision trees, while in RandF, the optimal number of leaves is decided by the user (often using grid search cross validation). As a result, a disadvantage of RandF is that there is more hyperparameter tuning necessary. Lastly, GB [25] can be considered as a similar model to ADA; both these models rely strongly on the aforementioned boosting principle. However, while ADA only uses a depth of one within its trees (i.e., learners), GB uses a depth of more than one. Typically, the depth is set to a maximum of 8 to 32 leaves.

*2.2. Features*

According to the literature, the features derived from heart rate are most valuable when predicting LOS [11–14]. This is, however, because heart rate is regulated by the autonomic nervous system and is highly influenced by immunologic and cardiovascular changes. Research in both adults and preterm infants indicates that low heart rate variability is often succeeded by a sepsis episode and occurs often before obvious clinical signs are recognized [26]. This abnormal pattern in heart rate serves as a potential marker or indicator of impending sepsis in these infants. The study suggests that monitoring heart rate characteristics could potentially aid in the early identification of sepsis, allowing for prompt intervention and treatment. Moreover, it has been shown that preterm infants present an overall transient deceleration in heart rate hours leading up to the diagnosis of sepsis [26]. Other variables such as blood pressure, temperature, respiratory rate, and oxygen saturation are considered to be less valuable when predicting sepsis. According to Sullivan et al. (2015) [14], this is due to several reasons. First, blood pressure is monitored by arterial catheters only at the beginning of the NICU stay and often not at the time during which the infant can develop sepsis. Second, temperature is not a trustful measure as

incubators automatically adjust environmental temperature to keep the infants temperate within normal range. Third, respiratory rate is often confounded by inaccuracy of the chest impedance signal. Lastly, acute changes in oxygen saturation have not yet been thoroughly studied for their association with sepsis.

While the impact of acute changes in oxygen saturation on sepsis is an area of interest, it has not yet been extensively investigated. Oxygen saturation levels can fluctuate rapidly during critical illness, including sepsis. However, the specific relationship between acute changes in oxygen saturation and the presence or progression of sepsis remains largely unexplored. Further research is needed to elucidate this association and understand the potential role of oxygen saturation as a clinical marker or predictor of sepsis, as studies have shown that an increased heart rate deceleration increased respiratory instability [17]. Furthermore, a study by Fairchild et al. (2017) [3] showed that a cross-correlation index between heart rate and oxygen saturation (SpO2) proved to be the best illness predictor for the preclinical detection of sepsis. Next, Sullivan et al. (2015) [14] indicated that heart rate is strongly modulated by respiration and blood pressure changes. This is in line with the study of Cabrera et al. (2021) [7], who found both respiration and heart rate variability features to be of the importance when predicting LOS using ML.

Lastly, the perfusion index (PFI) (i.e., the ratio of the pulsatile blood flow to the non-pulsatile static blood flow) is considered a useful parameter when predicting late onset sepsis [27]. Thus, although these features are less convincing according to the literature compared with HRV features, we considered them to be promising and included them in this study. Interestingly, to the best of our knowledge, no literature is available yet of the use of PFI and SpO2 as predictors of LOS in preterm infants in a machine learning context. In Table 2, we present an overview of the vital parameters used in the recent literature of studies involving LOS prediction using ML.

**Table 2.** Overview of vital parameters used by recent studies involving LOS prediction using ML. heart rate variability (HRV), respiratory frequency (RF), perfusion index (PFI), and oxygen saturation (SpO2).

|  | **LogR** | **NB** | **DT** | **RandF** |
|---|---|---|---|---|
| Leon et al. (2020) [8] | X |  | X | X |
| Joshi et al. (2020) [17] | X |  |  |  |
| Cabrera et al. (2021) [7] | X | X |  |  |
| Gomez et al. * (2019) [9] | X | X | X | X |

* this work assessed EOS instead of LOS.

## 3. Methods

### 3.1. Patient Population

In total, 46 NICU patients from the Emma Children's Hospital Amsterdam UMC were de-identified and made available for this project, among which 15 were LOS patients and 31 were control patients. The LOS patients were diagnosed as such when there was an administration of antibiotics for that patient beyond the third day of life. Thus, this group also included infants that were merely found to be clinically suspected of LOS without a positive blood culture. Patients from which the blood samples were taken in the first 48 h of recording were removed as LOS typically occurs beyond the first 72 h of life. Similarly, control patients for whom the duration of antibiotic administration was less than 48 h were also removed. In total, seven patients were found to meet these exclusion criteria, among which were one LOS patient and six control patients. Therefore, a cohort of 39 patients were considered suitable for the study.

The population characteristics are shown in Table 3. The categorical variables are presented as the number of cases and the corresponding percentages. The continuous variables are presented as the median and interquartile range. For categorical variables, we performed a significance test using the Chi-squared test, and for continuous variables we used the Mann−Whitney U test. Similar to the study of Leon et al. (2020) [8], the

population characteristics showed that the LOS patients were significantly more premature than the control group. Moreover, the LOS group weighed significantly less than the control group. LOS occurred typically between the 8th and 22nd day of life, with a median value of 15 days.

**Table 3.** Study population characteristics.

|  | LOS | Control | Sign. |
|---|---|---|---|
| n | 14 | 25 |  |
| Birthweight (gram) | 1140 (770–1700) | 1695 (1142–3290) | * |
| Apgar 1 min | 6 (4–7) | 7 (5–8) | N.S. |
| Apgar 5 min | 8 (7–9) | 8 (7–9) | N.S. |
| Apgar 10 min | 8 (8–8) | 8 (8–9) | N.S. |
| Gestational age (weeks) | 28 (25–32) | 31 (29–38) | * |
| Female | 11 (23.9%) | 4 (8.7%) | N.S. |
| Twins | 4 (8.7%) | 3 (6.5%) | N.S. |
| Died | 2 (14.3%) | 3 (12%) | N.S. |
| Age at start antibiotics (days) | 15 (8–22) |  |  |

* $p < 0.05$, N.S. Non-significant.

The sample size of the study population could also play a role in the differences in $p$-values. If there is a relatively small number of infants with extreme birth weights (e.g., extremely low birth weight or extremely high birth weight) in this study, the statistical power to detect an association with sepsis may be reduced. This could result in a higher $p$-value for birth weight compared with gestational age if the latter has a more evenly distributed range within the sample. However, if there are differences in the distribution between birth weight and gestational age groups, the $p$-values may differ accordingly, as premature birth can have implications on the baby's immune system development and susceptibility to infections. As these factors are biologically distinct, they may have different effects on the likelihood of developing sepsis. This could result in a higher $p$-value for birth weight compared to gestational age if the latter has a more evenly distributed range within the sample.

*3.2. Signal Processing*

For each infant, high resolution time series data of the continuous vital parameters (e.g., heart rate, perfusion index, oxygen saturation, and respiratory rate) were recorded using the Philips MPG90 monitor. After extraction from the monitor, all of the raw vital parameter data were stored in a data warehouse. Within each segment, we discarded all null (i.e., '0') and missing values. To derive features, the data were segmented using a 30-min sliding window, with no overlap. From each of these resulting time segments, a plethora of features were calculated for each vital parameter, as described in Table A1 in the Appendix A. The HRV features were calculated using the AURA healthcare API [28]. The features from the other vital parameters were calculated using the SciPy API [29].

*3.3. Feature Generation*

For each vital parameter (i.e., Heart frequency, respiratory frequency, perfusion index, and oxygen saturation), a plethora of features were calculated (see Table A1 in the Appendix A for an overview). In this section, each feature set is briefly described.

As mentioned, features derived from heart frequency are considered to have the most predictive value. These features are known as heart rate variability (HRV) features. All HRV features were derived from the interbeat intervals, also known as the normal-to-normal (nn) intervals. To quantify heart rate variability, roughly three types of features can be extracted from the nn-intervals.

### 3.3.1. Time-Domain Measurements

These parameters were calculated by deriving several statistics from the inter heartbeat intervals, known as the normal-to-normal intervals. These intervals are defined as the time between each successive heartbeat. Metrics were derived using the mean (mean_nn), the standard deviation, the root means square, the minimum (min_nn), maximum (max_nn), and the differences between the minimum and maximum (range_nn).

### 3.3.2. Frequency-Domain Measurements

In the frequency domain, we calculated the features that were derived from the frequency range bands of each segment, which reflected various ranges: very low frequency (vLF), low frequency (LF), and high frequency (HF). For instance, LF is derived from the spectral power density with a frequency range band of 0.04–0.15 Hz. The HF feature is derived from the range band of 0.15–0.40 Hz. Interestingly, it has been shown that LF is related to the sympathetic nervous system, whereas HF reflects the parasympathetic nervous system [6]. As a result, the LF/HF ratio can be calculated, which provides insight into the amount of sympathetic and parasympathetic activity.

### 3.3.3. Non-Linear Measurements

The non-linear parameters include sample skewness and entropy, which estimate the level of regularity and predictability of the signal, respectively. Moreover, we can calculate the measures from the Poincare plot. These measures reflect short- and long-term variability. Similar to the HF feature within the frequency domain, the Poincare plot reflects the parasympathetic nervous system in humans [30]. In addition, we considered the factors of respiratory saturation, oxygen saturation, and perfusion index. The features that were derived from the respiratory (RF), oxygen saturation (SpO2), and perfusion index (PFI) time series data were processed similarly as the time-domain and non-linear domain features previously mentioned. The features of the segments of these vital parameters were calculated every 30 min. The first group of features was similar to the time-domain features and included the standard deviation, mean, median, maximum, range, and maximum of each segment. The second group of features was similar to the above-mentioned non-linear measurements and included sample entropy and skewness.

### 3.4. Calibration Period

In order to assure the validity of our measurements, we calibrated each feature for each patient. To do so, we subtracted each feature using the median of the first 48 h of recording of the respective patient. We opted for this specific time window as Leon et al. (2020) [8] found that this window yields the best predictive performance for both linear and non-linear models. The use of such a period to calibrate the data is especially important as gestational age (i.e., maturity) and birth weight were found to significantly differ in the research population between the LOS and control group (see Table 3). It is important to account for these factors, as they are found to significantly impact the behavior of the parameters. For instance, Lange et al. (2005) [31] found that heart rate is significantly influenced by gestational age. By calibrating the data, we were able to control these differences.

### 3.5. Data Analysis

For the LOS group, all of the segments before $t_0$ were selected. $t_0$ is defined as the moment the blood samples were taken. Subsequently, all the segments of the LOS patients were labeled as infected. For the control group, we selected a moment at random beyond the third day of life as $t_0$ (i.e., corresponding to the definition of LOS), and labelled all the hours before $t_0$ as infected. For this study, we only considered the segments of 48 h before $t_0$; thus, all the segments that were present earlier than 48 h before $t_0$ were discarded. After data labelling and selection, we calibrated each feature with a calibration period of 48 h. To do so, we subtracted each feature with the median of the first 48 h of recording the respective patients. This resulted in a calibrated dataset (Δ Features). After the calibration

period, we normalized each feature using min−max scaling. Next, for each feature, we calculated the significance for time intervals starting from $t_i = -48$ h until $t_i = -6$ h, with 6 h increments using a Mann−Whitney U. We discarded a feature if the respective feature had one or more increments that were non-significant. We performed two methods to overcome co-linearity. In the first method, we dropped all the features with a significant correlation of 0.90 or higher from the calibrated feature set. This resulted in a dataset with only significant features (Δ Significant_Features), as an example shown in Figure 1. In the second method, we created a feature set using principal component analysis (PCA) on the calibrated features (Δ PCA_Significant_Features). We performed the PCA analysis in such a way that 95% of the variance in the feature set was retained. Thus, in total, two types of datasets were created: Δ Significant_Features and Δ PCA_Significant_Features.

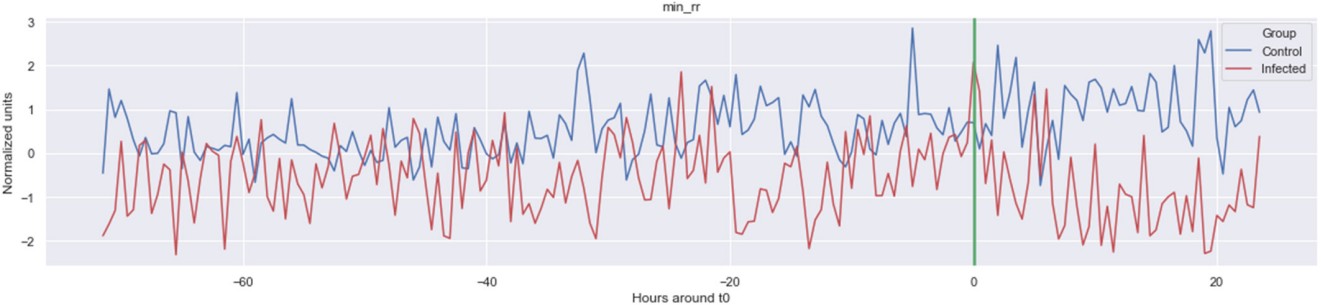

**Figure 1.** Example of one of the features calculated: minimum of inter heartbeat intervals (Δ Min_rr). The green line is the onset of the sepsis.

### 3.6. Machine Learning

In total, eight machine learning models were assessed: logistic regression (LogR), naive Bayes (NB), decision trees (DT), random forest (RandF), K-nearest neighbors (KNN), support vector machine (SVM), adaptive boosting (ADA), and gradient boosting (GB). As mentioned, LogR was considered as the baseline model as, in literature, it is generally considered to be the best performing model [7,8,17]. As the goal of the ML model was to either predict sepsis or LOS (i.e., "infected") for each 30 min segment starting from $t_i = -48$, all of the models could be considered binary classifiers. Similar to the approach of Gomez et al. (2019) [9], we used repeated stratified K-fold cross-validation (CV) to train and test the models. Repeated stratified K-fold is especially useful when dealing with imbalanced datasets, as it makes sure that each fold contains the same percentage of positive and negative labels as in the original study population. Thus, each fold contained roughly 36% LOS segments and 64% control segments. To avoid data leakage from the training set to the test set within the CV procedure, we ensured that no segments of the same patient were present in both the training and test set. As we used a four-fold CV procedure where each procedure was repeated 10 times, 40 test and train results were collected for each learning window. In order to ensure the reliability of the results, all of the results were presented using a 95% confidence interval. We considered the results of each model to be significantly different if the corresponding confidence intervals were non-overlapping. We assessed our models mainly on the predictive performance of the area under the receiving operator characteristic (AUROC) score. The AUROC score is suitable for imbalanced datasets, as it does not have any bias towards models that perform well on the minority class at the expense of the majority class [32]. Furthermore, we trained the models on a decreasing training window: each model was trained on data from a shorter period, starting at $t_i = -48$ and ending at $t_i = -6$, with 6 h increments. Because no further hyperparameter tuning was completed, all the data were used for the CV procedure.

We opted for no further hyperparameter tuning as this did not add much value towards the objective of this study (i.e., assess whether non-linear methods performed better than linear methods and on which types of features). All of the modelling was completed using the SciKit-learn library [24] within Python. The hyper-parameters as

advised by the documentation of this library were used. In Table A5 in the Appendix A, an overview of the cost functions is presented. For SVM, we used a "RBF" (i.e., radial basis function) kernel and for KNN the number of neighbors was set to 5 (i.e., the default parameters in the SciKit-learn library). In Table A6 in the Appendix A, an overview of the hyperparameters of the ensemble models is presented. Note that the "Max Estimators" and "Max Depth" refer to the amount of decision trees (i.e., "learners") and depth of the corresponding tree used in the ensemble models. However, if the ensemble reached a perfect fit with an amount less than specified within these parameters, the learning stopped early.

## 4. Results

In this section, we first present the general characteristics of each parameter before $t_0$ = (i.e., the moment when the blood samples were taken), by visually and statistically inspecting the features from each vital parameter. Second, we report the prediction results of the various ML models on the different feature sets of each vital parameter.

### 4.1. Feature Significance and Co-Linearity

We calculated the significance for each feature for time intervals starting from $t_i = -48$ h until $t_i = -6$ h, at 6 h increments, using a Mann−Whitney U. In Figure 2, a heatmap of the significance of each HRV feature is highlighted and Figure A6 in the Appendix A presents a complete overview of the significance of the features from all of the vital parameters. Interestingly, the results indicate that the significance of each feature differed over time. For instance, Figure 2 shows that the median (median_nni) of the periods $t_i = -48$ till $t_i = -30$ are non-significant. Moreover, the plot shows that the entropy (entropy_nn) feature was not significant around $t_i = -42$ and $t_i = -36$. For the other vital parameters, see Figure A6, we also saw that some features were non-significant. In both PFI and SpO2, it was shown that the mean and median of these parameters were non-significant. Moreover, we saw that in respiratory rate (RF), the skewness (RF_skew) was non-significant. We opted to discard all features with segments that were entirely or partly non-significant.

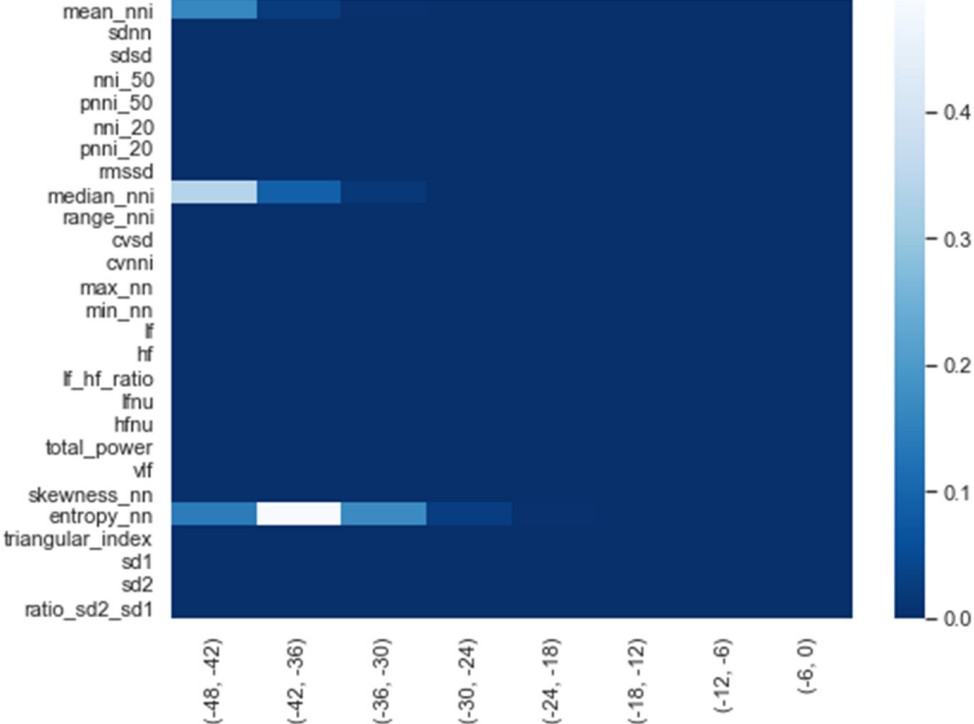

**Figure 2.** Significance of all HRV features over time from $t_i = -48$ h until $t_i = -6$ h, with 6 h increments.

After we selected the significant features, we performed a correlation analysis between the selected features for the entire time window (i.e., t = −48 h; see Figure 3). Interestingly, except for the PFI_mean and PFI_median features, only the HRV features were strongly correlated (i.e., correlation higher than 0.9). In order to avoid co-linearity, which would have inhibited us from drawing conclusions about the feature importance of the model, the co-linearity was removed. This is achieved by randomly selecting either one of each co-linear feature pair.

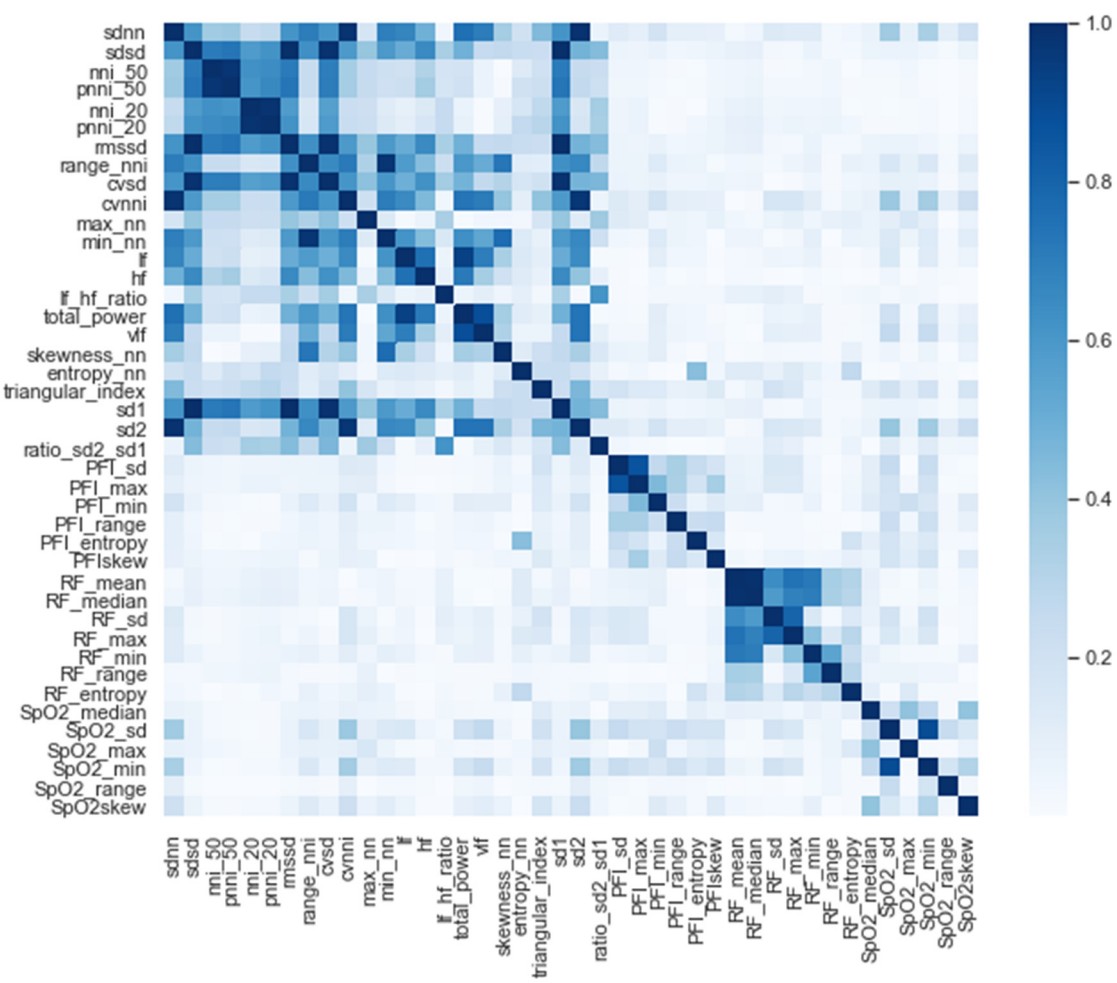

**Figure 3.** Correlation of all the significant features.

### 4.2. General Behavior of Each Parameter

In, Figures 1 and A1–A5, the mean of some significant features is shown over time for both the LOS and control patients.

The green line indicates the moment when the blood samples were taken. Interestingly, the figures clearly indicate that not all features are linearly separable and thus are of a non-linear nature. For instance, it was observed that the Δ nn_20 feature (see Figure A1) scored significantly higher for the infected patients than for the control patients for periods around $t_0 = -48$, but around $t_0 = -12$ this effect was reversed. Furthermore, for the Δ SpO2_median feature (Figure A4), it was observed that the infected group was significantly more volatile than the control group. Lastly, for both the Δ RF_median and Δ PFI_median (Figure A5), it was observed that the infected group had much more seasonality than the control group.

### 4.3. Predictive Performance of the Machine Learning Models

For each model, the area under the receiving operator characteristic (AUROC) was calculated. Moreover, we calculated the training time, accuracy ((TP + TN)/n), recall

(TP/(TP + FN)), and precision (TP/(TP + FP)), where TP = true positive, FN = false negative, FP = false positive. In Table 4, the results for each ML model for the entire training window are shown $t_i = -48$). In addition, Figure 4 shows the corresponding ROC curves for each model. Using the entire training window ($t_i = -48$), we found that all non-linear models performed superior compared with the linear models. In particular, for each metric within this training window, we found that RandF performed the best (AUROC score: 0.973) and NB (AUROC score: 0.734) performed the worst (except for training time). Furthermore, all models performed best on the normal dataset without any dimension reduction (i.e., Δ Significant_Features), Figure A9.

**Table 4.** Predictive performance of each model for the entire training window $t_i = -48$. The results are presented with a 95% Confidence Interval.

| | Fit Time | Train Accuracy | Test Accuracy | Train Precision | Test Precision | Train Recall | Test Recall | Train AUROC | Test AUROC |
|---|---|---|---|---|---|---|---|---|---|
| ADA | 0.519 [0.516, 0.521] | 0.899 [0.897, 0.9] | 0.865 [0.861, 0.868] | 0.899 [0.897, 0.9] | 0.864 [0.86, 0.868] | 0.899 [0.897, 0.9] | 0.865 [0.861, 0.868] | 0.966 [0.966, 0.967] | 0.936 [0.933, 0.939] |
| DT | 0.109 [0.107, 0.11] | 1.0 [1.0, 1.0] | 0.842 [0.839, 0.846] | 1.0 [1.0, 1.0] | 0.843 [0.839, 0.847] | 1.0 [1.0, 1.0] | 0.842 [0.839, 0.846] | 1.0 [1.0, 1.0] | 0.831 [0.827, 0.836] |
| GB | 2.249 [2.236, 2.262] | 0.942 [0.941, 0.943] | 0.899 [0.896, 0.902] | 0.944 [0.943, 0.945] | 0.902 [0.899, 0.905] | 0.942 [0.941, 0.943] | 0.899 [0.896, 0.902] | 0.988 [0.988, 0.989] | 0.963 [0.961, 0.965] |
| KNN | 0.023 [0.023, 0.023] | 0.937 [0.935, 0.938] | 0.898 [0.895, 0.9] | 0.937 [0.936, 0.938] | 0.898 [0.896, 0.9] | 0.937 [0.935, 0.938] | 0.898 [0.895, 0.9] | 0.985 [0.984, 0.985] | 0.95 [0.948, 0.952] |
| LogR | 0.021 [0.019, 0.022] | 0.748 [0.747, 0.749] | 0.743 [0.74, 0.746] | 0.744 [0.743, 0.745] | 0.739 [0.735, 0.742] | 0.748 [0.747, 0.749] | 0.743 [0.74, 0.746] | 0.794 [0.793, 0.795] | 0.782 [0.779, 0.786] |
| NB | **0.005** **[0.005, 0.005]** | 0.702 [0.699, 0.704] | 0.7 [0.694, 0.705] | 0.702 [0.7, 0.705] | 0.699 [0.693, 0.705] | 0.702 [0.699, 0.704] | 0.7 [0.694, 0.705] | 0.742 [0.74, 0.744] | 0.734 [0.729, 0.74] |
| RandF | 1.145 [1.139, 1.152] | 1.0 [1.0, 1.0] | **0.919** **[0.916, 0.922]** | 1.0 [1.0, 1.0] | **0.922** **[0.919, 0.924]** | 1.0 [1.0, 1.0] | **0.919** **[0.916, 0.922]** | 1.0 [1.0, 1.0] | **0.973** **[0.972, 0.975]** |
| SVM | 1.220 [1.216, 1.225] | 0.921 [0.92, 0.922] | 0.891 [0.888, 0.894] | 0.924 [0.923, 0.925] | 0.894 [0.891, 0.897] | 0.921 [0.92, 0.922] | 0.891 [0.888, 0.894] | 0.974 [0.974, 0.975] | 0.951 [0.949, 0.954] |

Bold is best performance.

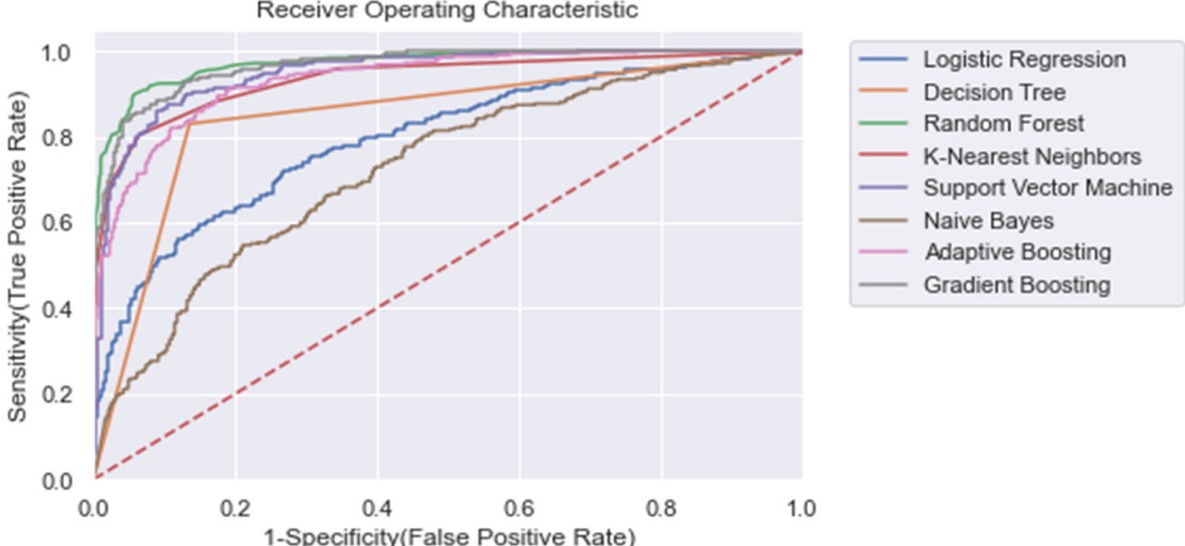

**Figure 4.** ROC plot of the models trained on the entire learning window (i.e., $t_i = -48$). Dashed line: Curve of a random classifier (the diagonal line).

The results also show that relatively adequate results were achieved using the conventional ML models, KNN and SVM, with both yielding an AUROC score of 0.950 and 0.951, respectively, which was comparable to the best performing ensemble models (i.e., GB and RandF). Interestingly, these models even outperformed ADA when trained on a longer time window ($t_i = -48$ to $t_i = -24$; see Figure 5 and Table 5). The strongest overfitting model was found to be the DT model, yielding a 17% lower AUROC score for the test than the training set. The least overfitting models were found to be LogR and NB, yielding

an AUROC score of 1% lower on the test set than on the training set. Except for DT, the non-linear models all scored an AUROC score between 2% and 3% lower for the test.

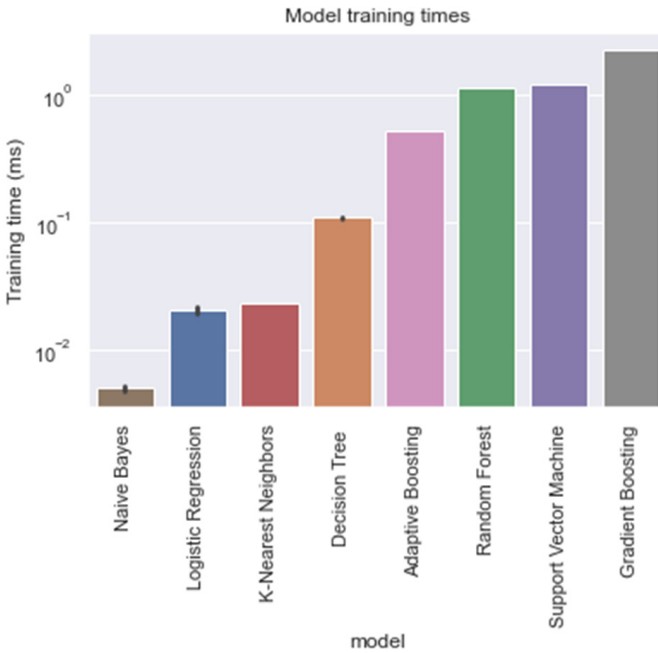

**Figure 5.** Training times of each model, trained on the entire training window (i.e., $t_i = -48$).

**Table 5.** AUROC test scores per training window. The results are shown with a 95% confidence interval.

| | −48 | −42 | −36 | −30 | −24 | −18 | −12 | −6 |
|---|---|---|---|---|---|---|---|---|
| ADA | 0.936 [0.933, 0.939] | 0.939 [0.936, 0.941] | 0.940 [0.938, 0.943] | 0.938 [0.936, 0.941] | 0.935 [0.932, 0.938] | 0.931 [0.926, 0.935] | 0.931 [0.925, 0.936] | 0.934 [0.926, 0.943] |
| DT | 0.831 [0.827, 0.836] | 0.827 [0.823, 0.831] | 0.817 [0.813, 0.822] | 0.808 [0.802, 0.813] | 0.805 [0.798, 0.813] | 0.796 [0.786, 0.805] | 0.784 [0.775, 0.793] | 0.792 [0.778, 0.806] |
| GB | 0.963 [0.961, 0.965] | 0.963 [0.962, 0.965] | 0.963 [0.961, 0.965] | 0.960 [0.958, 0.962] | **0.959** **[0.956, 0.962]** | **0.953** **[0.949, 0.956]** | **0.948** **[0.943, 0.952]** | **0.957** **[0.950, 0.965]** |
| KNN | 0.950 [0.948, 0.952] | 0.947 [0.945, 0.949] | 0.948 [0.946, 0.951] | 0.942 [0.939, 0.945] | 0.939 [0.935, 0.943] | 0.926 [0.921, 0.931] | 0.918 [0.912, 0.924] | 0.928 [0.921, 0.935] |
| LogR | 0.782 [0.779, 0.786] | 0.793 [0.788, 0.797] | 0.810 [0.806, 0.814] | 0.818 [0.813, 0.823] | 0.821 [0.816, 0.827] | 0.822 [0.813, 0.830] | 0.858 [0.850, 0.865] | 0.915 [0.908, 0.922] |
| NB | 0.734 [0.729, 0.740] | 0.737 [0.731, 0.742] | 0.742 [0.737, 0.747] | 0.746 [0.737, 0.754] | 0.750 [0.743, 0.758] | 0.764 [0.757, 0.771] | 0.792 [0.784, 0.801] | 0.810 [0.796, 0.824] |
| RandF | **0.973** **[0.972, 0.975]** | **0.971** **[0.970, 0.973]** | **0.970** **[0.968, 0.972]** | **0.966** **[0.964, 0.968]** | **0.964** **[0.961, 0.966]** | **0.955** **[0.952, 0.959]** | **0.947** **[0.943, 0.952]** | 0.956 [0.948, 0.963] |
| SVM | 0.951 [0.949, 0.954] | 0.948 [0.947, 0.95] | 0.951 [0.949, 0.954] | 0.948 [0.945, 0.951] | 0.951 [0.948, 0.954] | 0.943 [0.939, 0.947] | 0.944 [0.939, 0.949] | **0.962** **[0.955, 0.969]** |

Bold is best performance.

With respect to the training times of the models, RandF turned out to be a less beneficial model. As indicated in Table 4 and Figure 5, we observed that RandF (1.145 ms) turned out to need roughly two times more training time than ADA (0.519 ms). GB turned out to need the most training time (2.249 ms). Moreover, we found that the SVM model also took a relatively long time to be trained (1.220 ms). If we take both training and predictive performance into account, KNN was shown to be the best overall performing model (training time: 0.023 ms; AUROC: 0.950). In Table A3 in the Appendix A, we present the predictive performance of the PCA variant of the dataset. As observed, all models performed worse using this type of dataset. Interestingly, however, when inspecting the relative scores of the models using this dataset, KNN (AUROC: 0.938) and SVM (AUROC: 0.929) performed best compared with the ensemble models RandF (AUROC: 0.921), ADA (AUROC: 0.806), and GB (AUROC: 0.884).

Lastly, as we were dealing with an imbalanced dataset (i.e., 36% LOS and 64% control), we expected that the number of false positives would be bigger than the number of false

negatives. Nevertheless, the results show that the precision and recall scores for each model performed equally well.

### 4.4. Model Feature Importance

In Figure 6, we present the feature importance of the RandF model. Interestingly and in line with the literature [9], we observed that the HRV features were found to be significantly important. However, we found that the most important feature did not originate from the HRV feature set, but from the RF feature set (i.e., RF_mean). Furthermore, it was observed that perfusion index (PFI) and oxygen saturation (SpO2) were the least valued features.

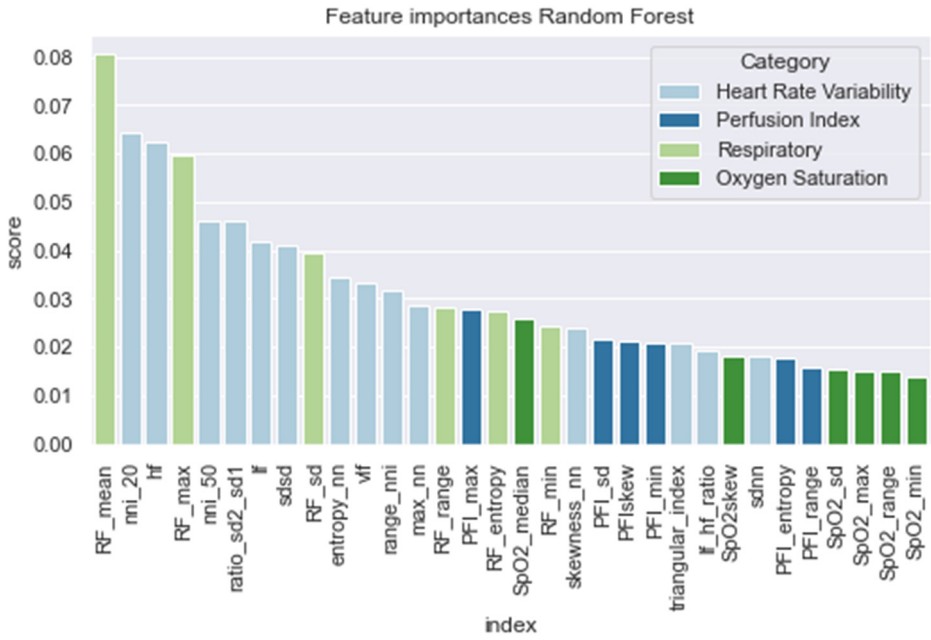

**Figure 6.** Feature importance of the random forest model.

### 4.5. Results for Different Training Windows

Figure 7 and Table 5 present the AUROC scores for the test set for a decreasing training window at 6 h increments. Overall, it was shown that RandF performed best for the training time $t_0 = -48$ to $t_0 = -36$. As a result of overlapping confidence intervals, it was observed that after this period, both GB and RandF performed equally well. Interestingly, due to overlapping confidence intervals, it was observed that for the training window $t_0 = -6$, besides RandF and GB, SVM performed similarly well.

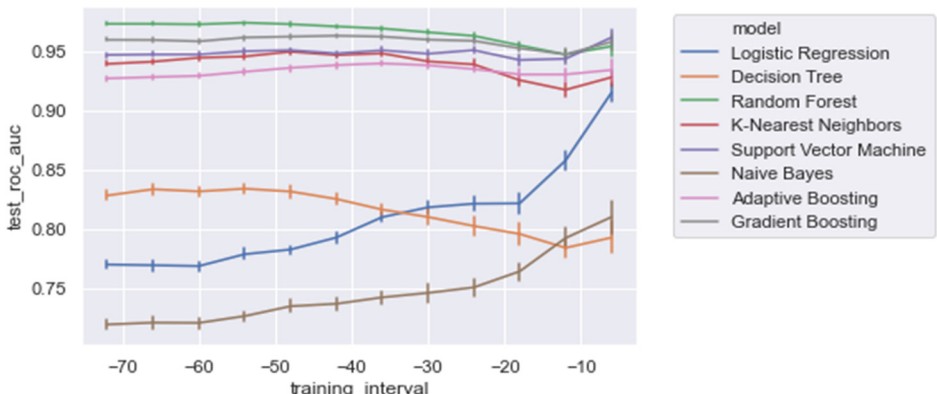

**Figure 7.** AUROC scores trained on a decreasing time window, trained on all of the significant features.

Moreover, it was shown that the linear models (i.e., NB and LogR) performed best for the shortest training window (i.e., $t_i = -6$). In particular, it was shown that LogR performed similarly well when compared with the non-linear methods when trained using $t_i = -6$ (AUROC: 0.915). In contrast, except for DT, the non-linear models were relatively stable for each training window increment (i.e., $t_i = -48$ to $ti = -6$), with generally slightly better performances for the longest training window (i.e., $t_i = -48$). Lastly, for all of the training windows, we found that all models performed better on the regular dataset (i.e., Δ Significant_Features) compared with the PCA variant of the dataset (i.e., Δ PCA_Significant_Features), see Figure A1 in the Appendix A.

*4.6. Performance of Vital Parameters*

In order to assess the value of each distinct vital parameter (i.e., HRV, RF, SpO2, and PFI) for predicting LOS, we trained each model on each vital parameter separately. In Figure 8, we present the AUROC test scores for all of the training windows (i.e., $t_i = -48$ to $t_i = -6$) for each feature set. As expected, and in line with the literature, the HRV features performed best. However, we also found that the ensemble models (i.e., RandF, ADA, and GB) yielded acceptable results for the feature sets of the parameters SpO2 and RF. In particular, using a training interval from around $t_i = -48$, we found that the models using features from SpO2 outperformed the RF features. On the other hand, we found that while all separate feature sets performed worst around $t_i = -6$, RF performed best around this time interval. Thus, this is an indication that the vital parameters of respiratory (RF) are best used shortly before $t_0$.

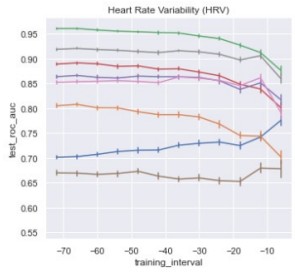 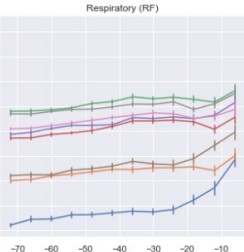 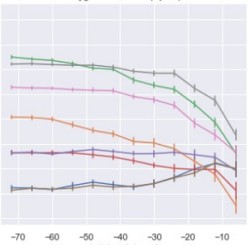 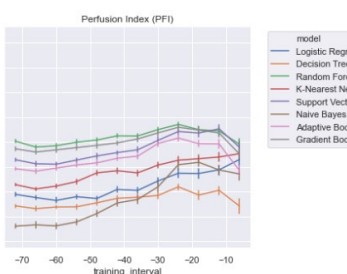

**Figure 8.** AUROC scores trained on a decreasing time window, trained on the distinct feature sets only.

All in all, when training the model with HRV features and adding the feature sets from RF or SpO2, it was observed that most models improved most when adding RF (see Table A2 and Figure A7 in the Appendix A). Interestingly, however, it was also observed that the GB model was most improved by adding features from SpO2. Thus, except for the GB model, we found that the models improved most when adding features from RF. Nevertheless, it is also important to mention that our best performing model (RandF) performed very well with the HRV feature alone. Lastly, it was observed that features from PFI added the least value to the model (see Table A2).

## 5. Discussion

In this study, the aim was to assess the added value of nonlinear methods and various vital parameters when predicting LOS in preterm infants. Ultimately, this study aimed to contribute to a new monitor that can be used as a non-invasive early warning system to alert medical professionals for an upcoming LOS episode. The main finding was that, in contrast with the findings of some works in the literature (e.g., [7,8]), non-linear models performed overall superior compared with linear models. The best performance was achieved by the RandF model using the largest training window (i.e., $t_i = -48$), achieving an AUROC score of 0.973, 24% better than the LogR model. In particular, the AUROC score of the RandF model was a 3% improvement compared with the best performing model of Gomez et al. (2021) [9]. Interestingly, the work of Gomez et al. (2021) [9] found that ADA performed

best compared with RandF. One reason for the underperformance of the ADA model might be the fact that ADA is typically more affected by noise (i.e., unimportant features) as it takes all features into account. In contrast, RandF only randomly selects a subset of the data during each training iteration. Generally, it was found that for most training windows, RandF performed best. However, when models were trained using the window $t_i = -6$, we found that GB and SVM were not performing significantly worse than the best performing model, RandF, due to overlapping confidence intervals. Furthermore, we found that the conventional machine learning models, SVM and KNN, were not significantly worse performing than the ensemble models—these models even outperformed ADA when trained on a longer time window (i.e., $t_i = -48$ to $t_i = -24$). Furthermore, we found that the linear models performed best when trained on a training window shortly before $t_0$, while the non-linear models were trained best on a longer training window. In particular, we found that, for a short training window, LogR performed similarly compared with some non-linear methods (e.g., KNN). Arguably, one reason for this behavior might be that linear methods performed better when trained shortly before $t_0$ because the data are less non-linear during this period. Moreover, we found that LogR was also the least overfitting model. Thus, considering these two findings, we still consider LogR as a promising model when trained shortly before $t_0$. The finding that non-linear models displayed a superior performance to linear models was in line with the observations regarding the non-linearity of several features. For instance, by inspecting the time series, we found that not all features were linearly separable; in Figure A1, it is observed that LOS patients did not score either higher or lower than the infected patients.

Furthermore, some features contained a strong seasonality (i.e., as observed in Figures A3 and A5). Hence, our finding that non-linear models performed better is explained by the non-linear behavior of the features. Although the GB model and RandF performed best with respect to the AUROC scores, the training times of this model were found to be less beneficial: these algorithms took 2.249 ms and 1.145 ms to run, respectively. Interestingly, KNN took significantly less time to train (i.e., 0.023 ms), but yielded a similar predictive performance (AUROC: 0.950). These training times are very important to take into consideration, as our models are expected to be deployed in a real-time setting, where models should be able to be trained in a timely manner. Therefore, it is important to take this trade-off between training time and predictive performance into consideration. As a result, we consider KNN to be a promising model. Interestingly, this is the first study that has reported such favorable results for the KNN model within the context of LOS prediction of neonates. Both Leon et al. (2020) [8] and Gomez et al. (2019) [9] reported that KNN was among the worst performing models in terms of AUROC score. Furthermore, it is worthwhile mentioning that this is the first study that reports the training times of the models within the context of LOS prediction using ML.

As expected, and in line with the literature, we found that all models performed best when using features derived from HRV. In contrast, the models performed worse than those trained on PFI features. For a training interval around $t_i = -48$, we found that ensemble models trained on SpO2 features also performed relatively well. However, when looking at the feature importance (i.e., Figure 6), we found that features derived from SpO2 were considered least important. Therefore, to overcome this ambiguity, future research is needed to discover the true effectiveness of the usage of features derived from SpO2 when predicting LOS. Interestingly, no papers have investigated the added value of features derived from SpO2 yet in the context of LOS prediction using ML. In addition, we found that while all separate vital parameter feature sets performed the worst around $t_i = -6$, RF (respiratory frequency) performed best around this time interval. This suggests that just before a sepsis event, RF might be a promising parameter to implement when predicting the risk of LOS shortly before the episode. In line with this finding, we found that adding features from RF to the HRV feature set added the largest increase in AUROC score for most models as in Table 4 and Figure A8. In contrast, adding PFI to the dataset effectuated the smallest increase in AUROC. Lastly, contradictory to the findings of Leon

et al. (2020) [8], we found that all models performed best on the feature set without any dimension reduction techniques applied (i.e., using PCA analysis). One reason for this finding might be that PCA, although it does remove noise from the dataset, does not necessarily lead to the selection of the most informative features [33].

## 6. Limitations and Future Work

During this study, several limitations were encountered. First, a small sample size was used to train the algorithms. Despite an imbalance in the number of patients per groups in our study, the incidence of LOS in this high risk population (14/39 patients = 36%) has been reported in number of other papers; thus, we feel that it is not likely to influence the current results as the current analysis does not predict whether an infant has a high risk of getting a LOS, but if vital parameters can predict LOS at an early stage and which technique (linear or non-linear) is superior to detect a LOS. For instance, Leon et al. (2020) used a sample size of 49 patients and Gomez et al. (2019) [9] used a sample size of 79 patients. Second, in the same vein as the study of Leon et al. (2020) [8], we found significant differences for gestational age and birthweight. Although we attempted to correct for these significant features using a calibration period of 48 h, it was not ruled out that these significant characteristics still influenced the predictive performance on the models. Third, although we aimed to overcome the class imbalance using the stratified version of K-fold cross-validation, it might still be the case that our algorithm was slightly biased towards the majority class (i.e., the control group). Nevertheless, Gomez et al. (2019) [9,11] also dealt with a similar class imbalance, so, from that perspective, our findings are externally validated. Fourth and related to the previous, it must be noted that our findings are best compared with the work of Gomez et al. (2019) [9], who used a similar cross-validation method (i.e., repeated stratified cross-validation).

On the contrary, Leon et al. (2020) [8] and Cabrera et al. (2021) [7] used leaving-one-out cross validation, which is known to yield more conservative results. Lastly, not all vital parameters and other LOS related factors that are available were assessed in this study; for instance, blood pressure, motion, clinical signs, and laboratory tests. Therefore, we encourage future scholars to also investigate the effect of other factors when predicting LOS using ML. A future direction is to explore the mean diagnosis of all AI algorithms including hybrid classifier-based particle swarm optimization (PSO) and neural network while keeping and preserving interpretability [34–36] for the early diagnosis of late onset sepsis.

## 7. Conclusions

In this study, we aimed to assess various linear and non-linear ML methods when predicting LOS. In addition, we assessed the added value of various vital parameters. In contrast with the findings in the literature, the results indicate that non-linear methods displayed a superior compared with the linear methods. Interestingly, we found that random forest performed best (AUROC: 0.973), without any dimension reduction (i.e., PCA) applied. However, when taking the trade-off between training time and predictive performance into account, we found that the K-nearest neighbors model was preferred when compared with the random forest. In line with the literature [37], we found that training the models on HRV features yielded the best results and adding RF (respiratory) features to the HRV features ensured the greatest improvement in terms of AUROC score for most models.

**Author Contributions:** Conceptualization, A.G.G.; methodology, A.G.G. and C.R.R.; software, A.G.G.; validation, A.G.G., C.R.R. and W.O.; formal analysis, A.G.G. and C.R.R.; investigation, A.G.G.; resources, A.G.G. and W.O.; data curation, A.G.G. and C.R.R.; writing—original draft preparation, A.G.G.; writing—review and editing, A.G.G. and C.R.R.; visualization, A.G.G.; supervision, C.R.R. and W.O.; project administration, C.R.R. and W.O.; funding acquisition, W.O. All authors have read and agreed to the published version of the manuscript.

**Funding:** This work was supported by the University of Amsterdam.

**Institutional Review Board Statement:** Written informed consent for participant and publication of the study results was obtained from all parents.

**Informed Consent Statement:** Written informed consent has been obtained from both parents of the patients to reuse the data for retrospective studies.

**Data Availability Statement:** Upon a reasonable request, the corresponding author can offer a partial code for the study upon completion of all projects.

**Conflicts of Interest:** The authors declare no conflict of interest.

## Appendix A

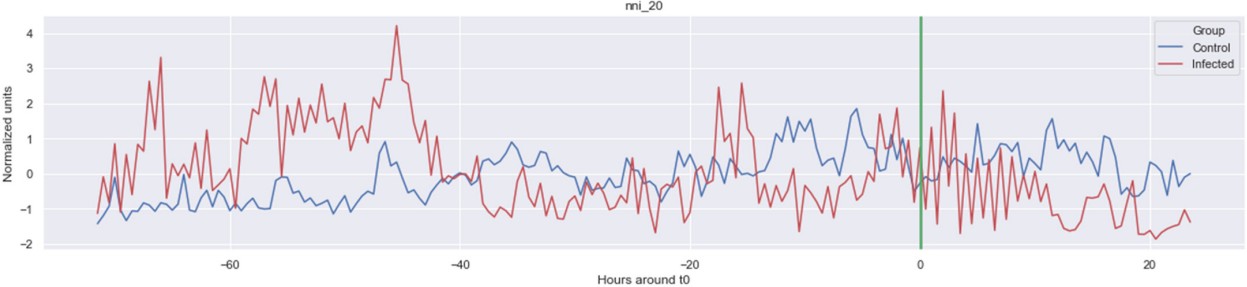

**Figure A1.** Δ nni. The green line is the onset of the sepsis.

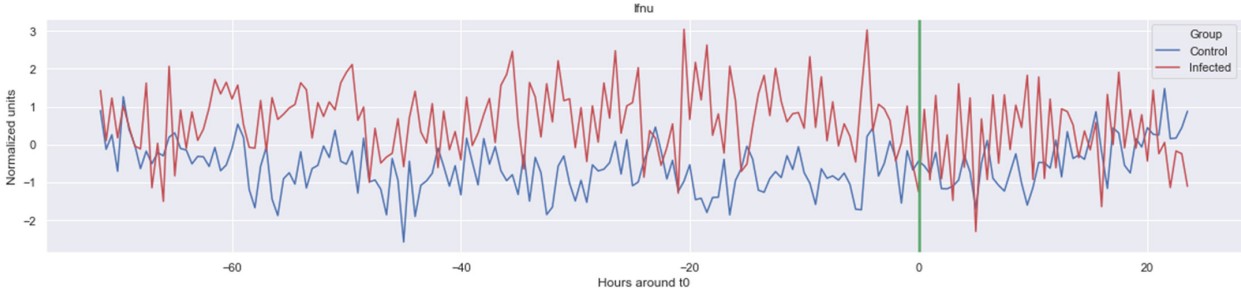

**Figure A2.** Δ lf. The green line is the onset of the sepsis.

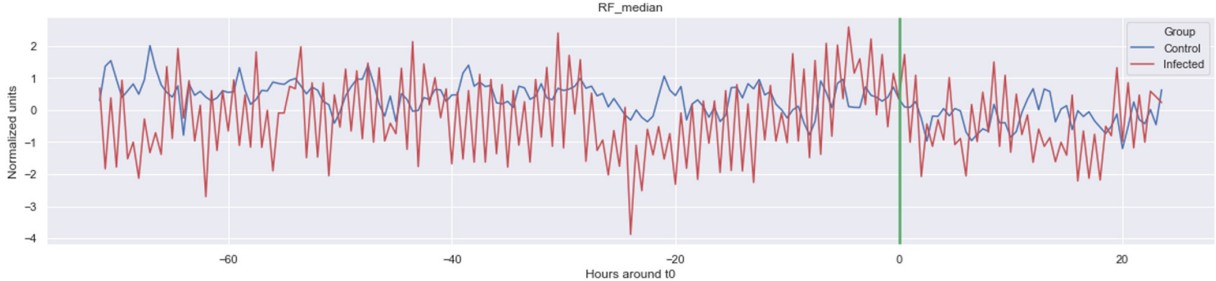

**Figure A3.** Δ RF_median. The green line is the onset of the sepsis.

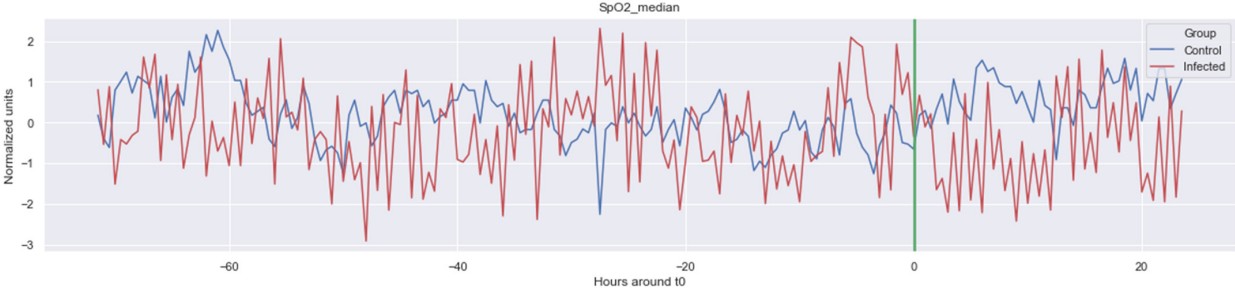

**Figure A4.** Δ SpO2_median. The green line is the onset of the sepsis.

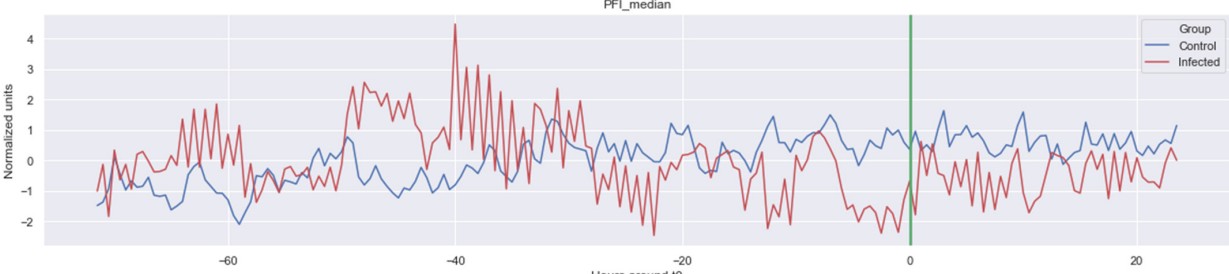

**Figure A5.** Δ PFI_median. The green line is the onset of the sepsis.

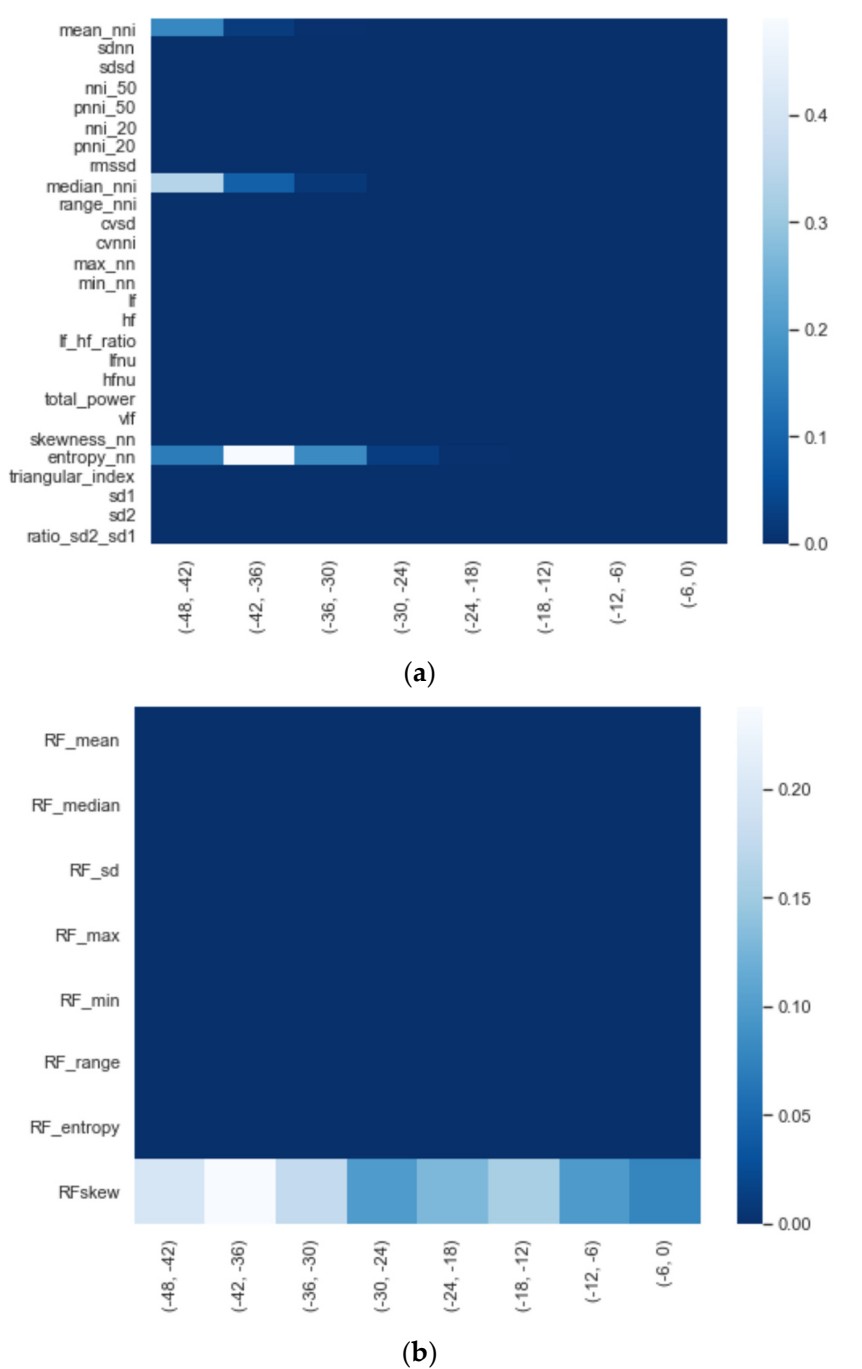

(a)

(b)

**Figure A6.** *Cont.*

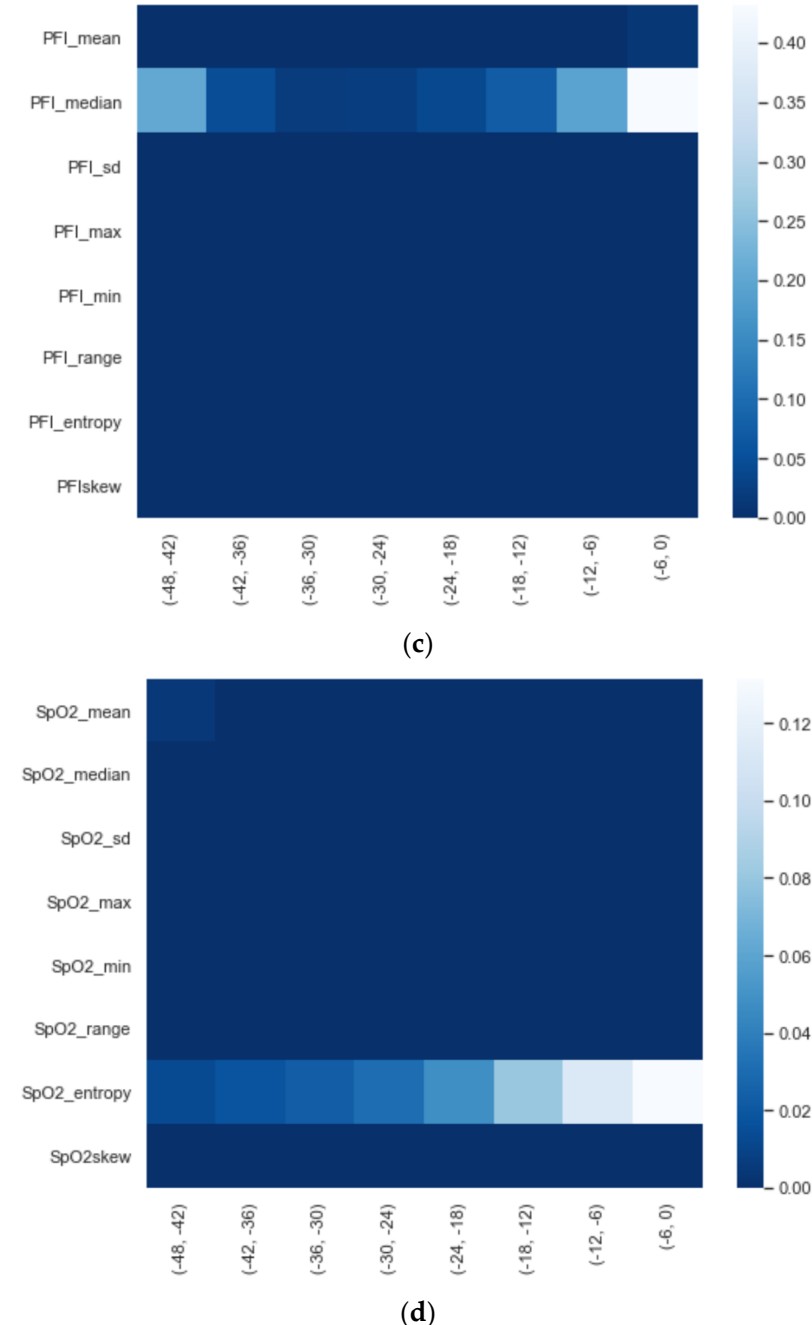

(c)

(d)

**Figure A6.** Significance of each set of features over time, starting from $t_i = -48$ to $t_i = -6$ with 6 h increments: (**a**) HRV, (**b**) RF, (**c**) PFI, and (**d**) SpO2.

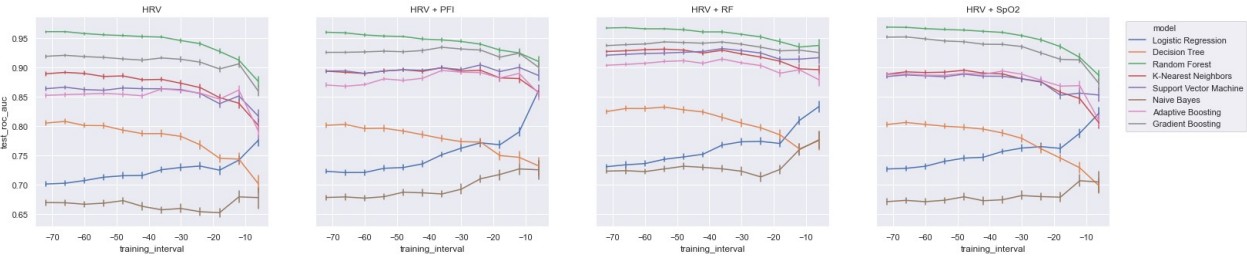

**Figure A7.** AUROC score for each combination of feature sets using a decreasing training window.

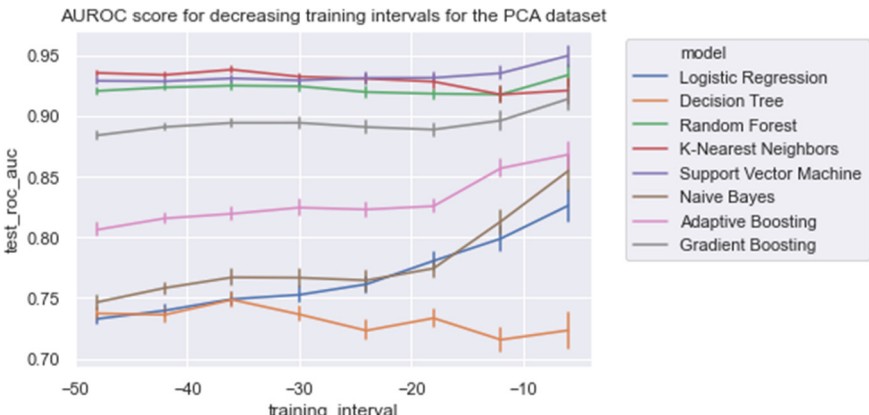

**Figure A8.** AUROC score for each combination of feature sets using a decreasing training window, trained on the PCA dataset (Δ PCA_Significant_Features).

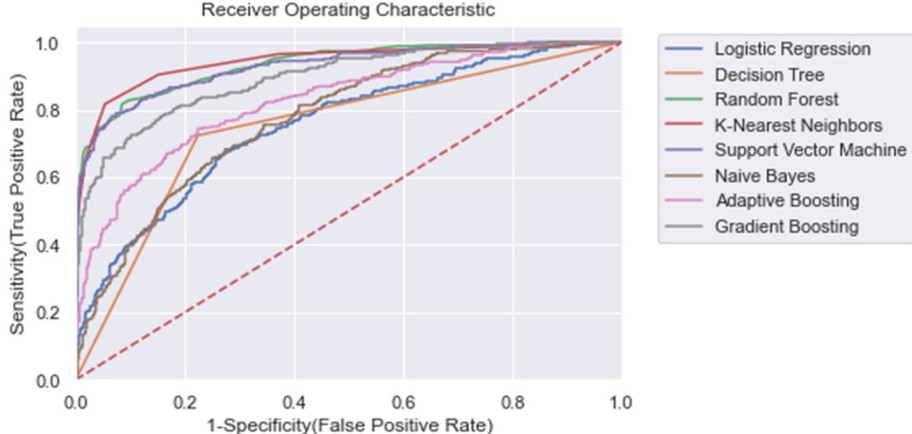

**Figure A9.** ROC plot of the models trained on the entire learning window, trained on the PCA dataset (Δ PCA_Significant_Features) (i.e., $t_i = -48$). Dashed line: Curve of a random classifier (the diagonal line).

**Table A1.** Overview of features that are extracted from the vital parameters. HRV: heart rate variability; RF: respiratory frequency, PFI: perfusion index, SpO2: oxygen saturation.

| Abbreviation | Unit | Interpretation |
|---|---|---|
| HRV | | |
| Time domain | | |
| mean_nni | ms | Mean of nni |
| sd_nn | ms | Standard deviation of nni |
| sd_diff_nn | ms | Standard deviation of differences between adjacent nn-intervals |
| rmssd | ms | The square root of the mean of the sum of the squares of differences between adjacent nni intervals |
| max_nn | ms | Maximum of nni |
| min_nn | ms | Minimum of nni |
| nni_50 | ms | Number of interval differences of successive nn-intervals greater than 50 ms |
| pnni_50 | % | The proportion derived by dividing nni-50 by the total number of nn-intervals |
| nni_20 | ms | Number of interval differences of successive nn-intervals greater than 20 ms |
| pnni_20 | % | The proportion derived by dividing nni20 by the total number of nni. |
| range_nni | ms | Difference between the maximum and minimum nn-interval. |
| cvsd | ms | Coefficient of variation of successive differences equal to the rmssd divided by mean_nni. |
| cvnni | ms | Coefficient of variation equal to the ratio of sdnn divided by mean_nni |
| Frequency domain | | |
| total_power | ms^2 | Total power density spectral |

**Table A1.** *Cont.*

| Abbreviation | Unit | Interpretation |
|---|---|---|
| vlf | msˆ2 | Variance in HRV in the Very low Frequency (0.003 to 0.04 Hz by default). Primarily modulated by sympathetic activity. |
| LF | msˆ2 | variance in HRV in the low Frequency (0.04 to 0.15 Hz). Reflects a mainly sympathetic activity |
| HF | msˆ2 | variance in HRV in the High Frequency (0.15 to 0.40 Hz by default). Reflects fast changes in HRV due to parasympathetic activity. |
| LF_HF_ratio | msˆ2 | Ratio lf/hf. This metric is as a quantitative mirror of the sympathetic/vagal balance. |
| LF norm | nu | normalized lf power(LF/(total powerVLF) $\times$ 100) |
| HF norm | nu | normalized hf power(hf/(total powerVLF) $\times$ 100) |
| csi | | Cardiac Sympathetic Index |
| cvi | | Cardiac Vagal (Parasympathetic) Index. |
| Non-linear domain | | |
| Entropy_RRi | | A measure of the degreee of distortion compared to a Gaussian distribution |
| Skewness_RRi | | A measure of the degreee of skeweness compared to a Gaussian distribution |
| triangular_index | | The triangular interpolation of RR-interval histogram is the baseline width of the distribution measured as a base of a triangle |
| sd1 | | sd1: The standard deviation of projection of the Poincare plot on the line perpendicular to the line of identity. |
| sd2 | | sd2: sd2 is defined as the standard deviation of the projection of the Poincare plot on the line of identity. |
| ratio_sd2_sd1 | | Ratio between SD2 and SD1. |
| RF | | |
| Time domain | | |
| Mean_RF | ms | Mean of the perfusion index |
| Median_RF | ms | Median of the perfusion index |
| SD_RF | ms | Standard deviation of the perfusion index |
| Max_RF | ms | Maximum perfusion index |
| Min_RF | ms | Minimum perfusion index |
| Range_RF | ms | Difference between the maximum and minimum RF-interval. |
| Non-linear domain | | |
| Entropy_RF | | A measure of the degree of distortion compared with a Gaussian distribution |
| Skewness_RF | | A measure of the degree of skewness compared with a Gaussian distribution |
| PFI | | |
| Time domain | | |
| Mean_PFI | % | Mean of the respiratory rate |
| Median_PFI | % | Median of the respiratory rate |
| SD_PF | % | Standard deviation of the respiratory rate |
| Max_PFI | % | Maximum respiratory rate |
| Min_PFI | % | Minimum respiratory rate |
| Range_PFI | ms | Difference between the maximum and minimum PFI interval. |
| Non-linear domain | | |
| Entropy_PFI | | A measure of the degree of distortion compared with a Gaussian distribution |
| Skewness_PFI | | A measure of the degree of skewness compared with a Gaussian distribution |
| SpO2 | | |
| Time domain | | |
| Mean_SpO2 | % | Mean of the saturation |
| Median_SpO2 | % | Median of the saturation |
| SD_SpO2 | % | Standard deviation of the saturation |
| Max_SpO2 | % | Maximum saturation |
| Min_SpO2 | % | Minimum saturation |
| Range_SpO2 | ms | Difference between the maximum and minimum SpO2-interval. |
| Non-linear domain | | |
| Entropy_SpO2 | | A measure of the degree of distortion compared to a Gaussian distribution |
| Skewness_SpO2 | | A measure of the degree of skewness compared to a Gaussian distribution |

**Table A2.** Models trained on the entire length (i.e, $t_i = -48$), but with separate combinations of features.

| | HRV | HRV + PFI | HRV + RF | HRV + SpO2 |
|---|---|---|---|---|
| Adaptive Boosting | 0.852 [0.849, 0.856] | 0.870 [ 0.867, 0.873] | **0.904 [0.901, 0.906]** | 0.888 [0.885, 0.891] |
| Decision Tree | 0.805 [0.801, 0.809] | 0.801 [0.798, 0.805] | **0.825 [0.821, 0.828]** | 0.803 [0.798, 0.807] |
| Gradient Boosting | 0.919 [0.916, 0.922] | 0.926 [0.923, 0.928] | 0.937 [0.935, 0.939] | **0.952 [0.950, 0.953]** |
| K-Nearest Neighbors | 0.889 [0.886, 0.892] | 0.894 [0.892, 0.895] | **0.927 [0.925, 0.929]** | 0.888 [0.885, 0.891] |
| Logistic Regression | 0.702 [0.698, 0.705] | 0.723 [0.719, 0.727] | **0.731 [0.728, 0.734]** | 0.727 [0.723, 0.731] |
| Naive Bayes | 0.670 [0.665, 0.675] | 0.679 [0.675, 0.683] | **0.723 [0.720, 0.727]** | 0.672 [0.667, 0.676] |
| Random Forest | 0.961 [0.959, 0.962] | 0.960 [0.958, 0.961] | **0.967 [0.966, 0.968]** | **0.969 [0.967, 0.970]** |
| Support Vector Machine | 0.864 [0.861, 0.867] | 0.894 [0.891, 0.896] | **0.921 [0.919, 0.923]** | 0.885 [0.881, 0.888] |

Bold is best performance.

**Table A3.** Predictive performance of each model for the entire training window $t_i = -48$ on the PCA dataset (Δ PCA_Significant_Features). The results are shown with a 95% confidence interval.

| | Fit Time | Train Accuracy | Test Accuracy | Train Precision | Test Precision | Train Recall | Test Recall | Train AUROC | Test AUROC |
|---|---|---|---|---|---|---|---|---|---|
| ADA | 0.431 [0.426, 0.436] | 0.804 [0.802, 0.806] | 0.759 [0.755, 0.763] | 0.803 [0.801, 0.805] | 0.755 [0.750, 0.76] | 0.804 [0.802, 0.806] | 0.759 [0.755, 0.763] | 0.872 [0.870, 0.873] | 0.806 [0.801, 0.811] |
| DT | 0.073 [0.07, 0.076] | 1.0 [1.0, 1.0] | 0.754 [0.750, 0.759] | 1.0 [1.0, 1.0] | 0.754 [0.750, 0.758] | 1.0 [1.0, 1.0] | 0.754 [0.750, 0.759] | 1.0 [1.0, 1.0] | 0.737 [0.733, 0.742] |
| GB | 1.732 [1.717, 1.748] | 0.891 [0.89, 0.892] | 0.823 [0.820, 0.826] | 0.897 [0.896, 0.898] | 0.826 [0.823, 0.829] | 0.891 [0.890, 0.892] | 0.823 [0.820, 0.826] | 0.960 [0.960, 0.961] | 0.884 [0.881, 0.887] |
| KNN | 0.019 [0.017, 0.021] | 0.925 [0.923, 0.926] | **0.880 [0.878, 0.882]** | 0.925 [0.924, 0.927] | 0.881 [0.878, 0.883] | 0.925 [0.923, 0.926] | **0.880 [0.878, 0.882]** | 0.980 [0.979, 0.980] | **0.935 [0.933, 0.938]** |
| LogR | 0.015 [0.014, 0.016] | 0.705 [0.703, 0.706] | 0.700 [0.697, 0.703] | 0.700 [0.699, 0.702] | 0.695 [0.691, 0.699] | 0.705 [0.703, 0.706] | 0.700 [0.697, 0.703] | 0.741 [0.740, 0.743] | 0.733 [0.729, 0.737] |
| NB | 0.005 [0.003, 0.008] | 0.701 [0.700, 0.702] | 0.698 [0.694, 0.703] | 0.695 [0.694, 0.697] | 0.692 [0.687, 0.697] | 0.701 [0.700, 0.702] | 0.698 [0.694, 0.703] | 0.751 [0.749, 0.753] | 0.747 [0.741, 0.752] |
| RandF | 1.092 [1.081, 1.102] | 1.0 [1.0, 1.0] | 0.859 [0.856, 0.862] | 1.0 [1.0, 1.0] | **0.863 [0.860, 0.866]** | 1.0 [1.0,1.0] | 0.859 [0.856, 0.862] | 1.0 [1.0, 1.0] | 0.921 [0.918, 0.923] |
| SVM | 0.226 [0.223, 0.230] | 0.898 [0.897, 0.899] | 0.863 [0.860, 0.866] | 0.903 [0.902, 0.904] | **0.867 [0.864, 0.87]** | 0.898 [0.897, 0.899] | 0.863 [0.860, 0.866] | 0.960 [0.959, 0.960] | 0.929 [0.926, 0.931] |

**Table A4.** AUROC test scores per training window on the PCA dataset (Δ PCA_Significant_Features). The results are shown with a 95% confidence interval.

| | −48 | −42 | −36 | −30 | −24 | −18 | −12 | −6 |
|---|---|---|---|---|---|---|---|---|
| ADA | 0.806 [0.801, 0.811] | 0.816 [0.811, 0.82] | 0.819 [0.815, 0.824] | 0.824 [0.819, 0.83] | 0.823 [0.817, 0.829] | 0.826 [0.82, 0.831] | 0.857 [0.849, 0.865] | 0.868 [0.857, 0.879] |
| DT | 0.737 [0.733, 0.742] | 0.736 [0.730, 0.742] | 0.749 [0.743, 0.755] | 0.737 [0.731, 0.742] | 0.723 [0.715, 0.731] | 0.733 [0.727, 0.740] | 0.716 [0.706, 0.726] | 0.723 [0.709, 0.738] |
| GB | 0.884 [0.881, 0.887] | 0.891 [0.888, 0.894] | 0.894 [0.891, 0.898] | 0.894 [0.890, 0.899] | 0.891 [0.886, 0.896] | 0.889 [0.884, 0.894] | 0.896 [0.889, 0.904] | 0.914 [0.904, 0.923] |
| KNN | **0.935 [0.933, 0.938]** | **0.934 [0.931, 0.936]** | **0.938 [0.935, 0.941]** | 0.932 [0.93, 0.935] | 0.931 [0.927, 0.935] | **0.928 [0.923, 0.933]** | 0.918 [0.911, 0.925] | 0.921 [0.911, 0.931] |
| LogR | 0.733 [0.729, 0.737] | 0.74 [0.735, 0.744] | 0.749 [0.744, 0.754] | 0.753 [0.746, 0.759] | 0.761 [0.755, 0.768] | 0.781 [0.773, 0.788] | 0.799 [0.788, 0.809] | 0.826 [0.813, 0.839] |
| NB | 0.747 [0.741, 0.752] | 0.758 [0.754, 0.763] | 0.767 [0.761, 0.773] | 0.767 [0.760, 0.774] | 0.765 [0.757, 0.773] | 0.774 [0.767, 0.782] | 0.813 [0.803, 0.823] | 0.854 [0.839, 0.87] |
| RandF | 0.921 [0.918, 0.923] | 0.923 [0.921, 0.926] | 0.925 [0.922, 0.928] | 0.924 [0.921, 0.928] | 0.920 [0.915, 0.924] | 0.918 [0.914, 0.923] | 0.918 [0.911, 0.924] | 0.934 [0.924, 0.943] |
| SVM | 0.929 [0.926, 0.931] | 0.928 [0.926, 0.931] | 0.931 [0.928, 0.934] | **0.929 [0.926, 0.932]** | 0.931 [0.927, 0.936] | **0.931 [0.927, 0.935]** | **0.935 [0.930, 0.941]** | **0.950 [0.941, 0.958]** |

**Table A5.** Overview of the cost function and number of estimators used in the models.

| Model | Cost Function | Estimators |
| --- | --- | --- |
| ADA | Exponential loss | 50 |
| DT | Gini impurity | - |
| GB | Log likelihood | 100 |
| KNN * | - | - |
| LogR | Cross entropy | - |
| NB | Negative point log-likelihood | - |
| RandF | Gini impurity | 100 |
| SVM | Hinge loss | - |

* K-Nearest Neighbor (KNN).

**Table A6.** Overview of the characteristics of the ensemble models.

| Model | Split Criterion | Learner | Max Estimators | Max Depth |
| --- | --- | --- | --- | --- |
| ADA | Exponential loss | Decision Tree | 50 | 1 |
| GB | MSE | Decision Tree | 100 | 3 |
| RandF | Gini impurity | Decision Tree | 100 | 1000 |

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
