# Peer review of "Early Detection of Late Onset Sepsis in Extremely Preterm Infants Using Machine Learning: Towards an Early Warning System"

_applsci, doi:10.3390/app13169049_

Round 1

Reviewer 1 Report (Previous Reviewer 2)

Dear authors,

thank you for the revision of your manuscript. All issues have been adressed, but the order of citation still needs correction. The references should be numbered according to their appearence in the text.

For example: 

29. Barbara J Stoll, Nellie Hansen, Avroy A Fanaroff, Linda L Wright, Waldemar A Carlo, Richard A Ehrenkranz, James A Lemons, 756 Edward F Donovan, Ann R Stark, Jon E Tyson, et al. 2002. Late-onset sepsis in very low birth weight neonates: the experience 757 of the NICHD Neonatal Research Network. Pediatrics 110, 2 (2002), 285–291.

This should be no. 1, since it is the first literature that is citated.

Minor misspellings could be corrected during the editorial process.

Author Response

Dear Reviewer,

attached please find our responses.

Thank you.

Reviewer 2 Report (Previous Reviewer 1)

1 I did not see any changes in yellow color.

2 Fig 1 legend missing !!

3 in Table 2 ,Heart Rate Variability (HRV), Respiratory Frequency (RF), Perfusion Index (PFI), Oxygen Saturation (SpO2). *: this work assessed EOS instead of LOS.should under table marked as note.

4 In Table 3. Study population characteristics,why Birthweight (gram) and Gestational age were shown significant difference(*: p < 0.05).the patients data were not different which showed data imbalance.

5 the reference is dizzy which not in sequence ,please meet journal requirement.

6 All Figs need improve quality of resolution.

Author Response

Dear Reviewer,

attached please find our responses.

Thank you.

Reviewer 3 Report (New Reviewer)

After reviewing a manuscript entitled "Early detection of Late Onset Sepsis in extremely preterm infants using machine learning: Towards an early warning system", we would like to inform you that this manuscript has good clinical value. It's well organized and covers the existing chat I've seen before. This study investigates and evaluates the effect of nonlinear methods and examines whether other vital parameters such as respiration, perfusion index and oxygen saturation can have added value when predicting LOS and also showed that nonlinear methods were superior compared to linear models.

This study has a good quality, but several steps can improve the quality of the article, which is necessary for the publication of this article.

1: In the abstract, briefly present some information about the conclusion (you did not mention this issue).

2: In terms of English writing, it needs a little editing.

3: To enrich the list of references, I suggest you use the following references. These references can increase the bibliographic value of your references.

"Efficient Model for Coronary Artery Disease Diagnosis: A Comparative Study of Several Machine Learning Algorithms"

"A hybrid particle swarm and neural network approach for detection of prostate cancer from benign hyperplasia of prostate"

"Management of covid-19 detection using artificial intelligence in 2020 pandemic"

All sections of this manuscript are well presented except for the limited cases mentioned in the revisions. I hope that by doing these revisions, this study will be given the right to be published in this journal.

Needs to be revised.

Author Response

Dear Reviewer,

attached please find our responses.

Thank you.

Reviewer 4 Report (New Reviewer)

Generally, the manuscript was well organized and the contents are interesting. It is suggested to accept this manuscript after minor revision.

Q1, the figure quality should be improved greatly

Q2, cite more relevant references to support the claim

good

Author Response

Dear Reviewer,

attached please find our responses.

Thank you.

Round 2

Reviewer 2 Report (Previous Reviewer 1)

  • 1 many reviewers pointed out that figs need be improved quality ,however they insisited on that" Quality cannot be improved and must be considered as-is. They are in good quality"
  • 2 I have stated that reference is not in sequence ,however he responsed that "All references were ordered by appearance following journal requirements." I showed that  following is the first apperance in ref section . (29) Barbara J Stoll, Nellie Hansen, Avroy A Fanaroff, Linda L Wright, Waldemar A Carlo, Richard A Ehrenkranz, James A 703 Lemons, Edward F Donovan, Ann R Stark, Jon E Tyson, et al. 2002. Late-onset sepsis in very low birth weight neonates: the 704 experience of the NICHD Neonatal Research Network. Pediatrics 110, 2 (2002), 285–291.
  • 3 He responseed that track change in yellow in first revision ,but I did not see in yellow ,this time he responsed that "New changes are highlighted in green."

Author Response

(The authors gave the same response as above.)

Round 3

Reviewer 2 Report (Previous Reviewer 1)

there is no change about response to my comment . SUGGEST complete my comment.

This manuscript is a resubmission of an earlier submission. The following is a list of the peer review reports and author responses from that submission.

Round 1

Reviewer 1 Report

1 The figs need be improved quality of resolution.In addition ,Fig 1 need legend to give more explaintion.

2 Table 1 need note "*“ and abbreviation ,and the references are very old .

3 NICU in the abstract need give full name when in the first time.

4 The limitations of sample size is clear ,how to you explain this issue about smal simple size.

5 46 NICU patients included 15 LOS and 31 CONTROL patients, How to over the bias of the different significantly distribution of patient groups?

Reviewer 2 Report

Dear authors,

thank you for the submission of your manuscript, entitled: <Early detection of LOS in extremely preterm infants using machine learning: Towards an early warning system>

As you clearly state, sepsis is a serious, and sometimes fatal complication in the NICU. Therefore a rapid, early, and specific "warning systems" would be highly desirable.

Your study design is well choosen and described, and the results are properly presented.

However, I still have one major comment:

Although from the results you present, obviously 2 non-linear ML methods (Random Forest and K-nearest neighbors) are superior to the widely used linear models. As is used for example in the HeRO monitors.

For specialists in informatics, your manuscript and results might be crystal-clear, however, the neonatologists in the NICUs might want to know, how can this be applied in the NICU. This has to be added into the discussion and clonclusions (even if there is not yet a new device at hand). Especially when the title says at the end: <... Towards an early warning system>

Without this explanations and discussions, I think the manuscript is not suitable for "applied sciences".

Minor comments:

1.) LOS should be written out in the title

2.) ML should be written out in the abstract 

3.) Literatur should be counted as the appear in the manuscript

4.) page 1, lines 43-45: "... , which is often costly time when treating sepsis."  Probably you mean that "..., it can take up to 48 hours before a result is available, and that means that valuable time is lost when treating sepsis."

5.) page 2, line 51; NEC should be spelled out

6.) page 2, line 64; and page 3, line104; (e.g. and bracket ")" should be deleted. [literature] is enough

7.) page 2, lines 92-64; the term "immaturity" is strange in the context of ML as novel method.  

8.) page 3, line 103; "tool monitoring tool" one tool to much

9.) page 3, lines118-119: "..., 8 machine learnig ??? will be assessed:" methods, tests ? 1 word is missing

10.) page 3, lines117-126; The abbreviation for Random Forest should be given in line 117, and then it should be used concistently either RandF, or RF

11.) In tables 1 and 2, the literatur number should be added after the citation

12.) page 4, lines 180-181; "Moreover, it is shown ..." this sentence is hardly understandable. Please write this more clearly.

13.) the same is true for "Desite these drawbacks, ..." page 5, line191-192

14.) page 8, line 323-324; "..., it is generally considered to BE the best ..." the "be" is missing

Reviewer 3 Report

Thank you for allowing us to review your research. 

Need a thorough check on Grammer.